



# Mechanism of Delayed Storm Surge in Straits: Seiche-Induced Oscillations Triggered by Typhoon Passage

Shinichiro Ozaki[1], Yoshihiko Ide[2], and Masaru Yamashiro[2]

[1]Department of Civil Engineering, Graduate School of Engineering, Kyushu University, Fukuoka, Japan
[2]Disaster Risk Reduction Research Center, Graduate School of Engineering, Kyushu University, Fukuoka, Japan

**Correspondence:** Shinichiro Ozaki (ozaki.shinichiro.247@s.kyushu-u.ac.jp)

**Abstract.** A storm surge is a phenomenon in which the sea level rises significantly due to low-pressure systems, such as typhoons, accompanied by strong winds. Once storm surge-induced flooding occurs, it can rapidly inundate low-lying areas. Generally, the primary contributor to storm surge is wind set-up, where wind forces the sea towards the coast. As such, it is well-known that severe storm surges occur at typhoon's closest approach because of strong wind set-up. However, when

Maysak (2020) struck the northern coast of Kyushu Island (NCKI), located on the south side of the Tsushima Strait, the sea level rose and flooding occurred approximately half a day after the typhoon had passed. At NCKI, both atmospheric pressure and wind had already weakened at the time of the flooding. Thus, the storm surge could not be explained by wind set-up or the inverted barometer effect. We examined storm surge observations for typhoons that impacted NCKI over the past 20 years and revealed a tendency for two peaks in storm surge when typhoons passed through the western channel of the strait. The second

peak was identified as the maximum storm surge height, occurring approximately 10 hours after the typhoon had passed. The first peak occurred when the typhoon was closest to NCKI, coinciding with the time of minimum atmospheric pressure. This was attributed to the sea level rise caused by the inverted barometer effect. After the first peak, oscillations with a period of approximately 10 hours were observed, resulting in the second peak. NCKI, located along the Tsushima Strait, is subject to the geographical characteristics of the strait, which likely caused the oscillations leading to the maximum storm surge. To identify

the oscillations that occurred after the typhoon's passage, a continuous wavelet transform was applied to the results of storm surge simulations for time-frequency analysis. As a result, it was found that two types of seiches in a two-dimensional spatial domain of the strait (5-hour and 10-hour periods) occurred after the typhoon's passage. These seiches were triggered by the release of potential energy as external forces weakened following the typhoon's transit through the strait. Furthermore, the seiches were observed to occur approximately two hours earlier when the external force was wind, compared to when it was

atmospheric pressure. This is because the time variation of atmospheric pressure drop is slower than that of wind direction. In this study, we identified the occurrence of anomalous storm surges caused by typhoons passing through a strait under specific conditions and conducted a detailed investigation of their generation mechanisms, and demonstrated storm surges can occur even after a typhoon has passed and improved understanding of storm surge characteristics in straits.





## 1 Introduction

Storm surges are phenomena where sea levels rise significantly due to low air pressure and strong winds, such as during a typhoon. Once a storm surge occurs, extensive flooding rapidly affects low-lying coastal areas. Climate projections that account for global warming suggest that storm surge heights will increase worldwide in the future (Balaguru et al. (2016), Yang et al. (2020), and Mori et al. (2022)). To mitigate damage caused by storm surges, it is crucial to implement both structural and

non-structural measures. A thorough understanding of the nature of storm surges is essential for devising effective strategies.

Several widely accepted theories explain the storm surges. One such theory is the inverted barometer effect, where sea levels rise as if drawn up by low air pressure. Other theories include wind set-up and wave set-up (Wunsch and Stammer (1997), Walton and Dean (2009), and Wu et al. (2018)). Wind set-up occurs when strong winds force seawater ashore and raising sea levels. Wave set-up is caused by wave breaking and radiation stress. These mechanisms are most active when typhoons are

nearest to the target area due to the strongest winds and lowest air pressure. Additionally, previous studies have identified other causes of storm surges, such as seiche in bays and on continental shelves, Ekman set-up where seawater is pushed to the shore by the Coriolis force from prolonged winds parallel to the shore, and the propagation of high water levels due to shelf waves (Kim et al. (2010), Shen and Gong (2009), and Kennedy et al. (2011)).

Among these theories, wind set-up is the most significant contribution to the storm surge height. Wind set-up height is

directly proportional to fetch and inversely proportional to water depth. Regions that have historically suffered significant storm surge damage are often bays facing the ocean, characterized by geographical conditions that amplify wind set-up (Bilskie et al. (2016), Bhaskaran et al. (2020), Nakajo et al. (2015), Ide et al. (2020)).

Conversely, even in straits with relatively short fetch, significant storm surges can happen as a direct result of winds. For example, in East Asia, Tsushima Strait, located between Kyushu Island and the Korean Peninsula (see Figure 1), is frequently

hit by typhoons. Typhoons approaching the Tsushima Strait often move northeastward through the strait, after which they tend to either make landfall on the southern coast of the Korean Peninsula (SCKP, Figure 1) or enter the Sea of Japan. Sanba (2012), the most powerful typhoon of 2012, moved through the Tsushima Strait. The typhoon was accompanied by strong winds with a maximum instantaneous wind speed of over 20 m/s, and over 100 cm added surge height (Yoon et al. (2014)).

In contrast, on the northern coast of Kyushu Island (NCKI, see Figure 1), located across the Tsushima Strait from SCKP,

large storm surges generally unlikely occur during typhoons approaching because winds does not force the sea towards the coast. However, when Maysak (2020) struck NCKI, extensive areas were inundated, leading to house flooding and traffic disruptions (Niimi et al. (2022)). Maysak passed through the western channel of the Tsushima Strait (WCTS, see Figure 1) and did not make landfall at NCKI. Remarkably, the inundation occurred approximately 10 h after the closest approach to NCKI, with the typhoon over 600 km away from NCKI. The storm surge cannot be attributed to the inverted barometer effect, wind

set-up, or wave set-up, because atmospheric pressure had risen to about 1000 hPa, and there were no strong winds at the time of flooding.

In research focused on storm surges in NCKI, Hong and Yoon (1992) investigated the storm surge mechanism for Holly (1984). They compared the observed storm surge at NCKI and SCKP, and reported that there is a time lag of approximately





half a day between the maximum peak of storm surge anomalies between NCKI and SCKP. Niimi et al. (2022) studied the

effect of Coriolis force to the storm surge at NCKI during Maysak (2020) and concluded that oscillations occur in the Tsushima Strait after the passage of the typhoon regardless of the presence of the Coriolis force, and the maximum surge height becomes larger when the Coriolis force is present.

As indicated by these previous studies, oscillations occur in the Tsushima Strait following the passage of a typhoon, and it is highly likely that these oscillations lead to storm surges with a delay of approximately half a day. Storm surge mechanisms

in other straits are described in Zhang et al. (2010), Soontiens et al. (2016) and Tkalich et al. (2013). In such straits, there is also a possibility that storm surges may occur at unexpected times after the passage of a typhoon.

Therefore, this study aims to elucidate the mechanism behind surge increases that occur after typhoon passage, with the Tsushima Strait as a representative case, using observational data and numerical simulations.

The structure of this paper is as follows: In Section 2, we outline the methods for analyzing observational data and provide

details of the numerical simulations. Section 3 explores the mechanisms of storm surge generation specific to NCKI. Finally, Section 4 summarizes the findings of this study.

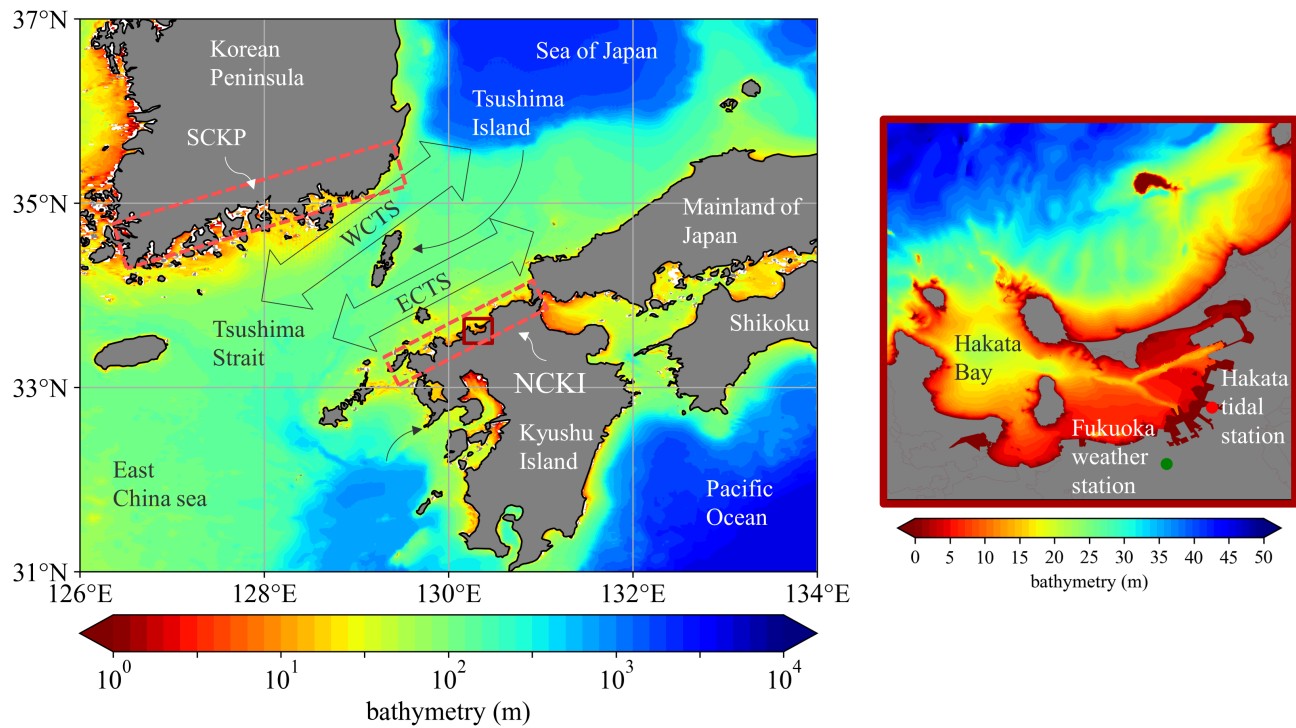

**Figure 1.** Bathymetry of the Tsushima Strait, with the points of the tidal stations around the Tsushima Strait, and the right side of the figure shows an enlarged view of Hakata Bay, with the points of the tidal station and the weather station.





## 2 Data and Methodology

### 2.1 Typhoons

We selected the typhoons used in this study as follows: First, we obtained data on 1,898 typhoon tracks from the best track
data provided by the Japan Meteorological Agency (JMA, https://www.jma.go.jp/jma/jma-eng/jma-center/rsmc-hp-pub-eg/
besttrack.html). Second, we identified 599 typhoons from 1998 to 2022 for which tide level observation data were available
for the Hakata tidal station. Finally, from these 599 typhoons, we selected 67 that passed within a 300 km radius of the Hakata
tidal station.

### 2.2 Storm surge anomalies

We calculated the storm surges caused by the 67 selected typhoons as follows: First, we obtained sea-surface elevation data
sampled at 30-s intervals from the Hakata tidal station for the period 1998 to 2022 via the RDMDB Data Retrieval System
(https://near-goos1.jodc.go.jp/vpage/search.html). Second, we performed a harmonic analysis on the obtained sea-surface el-
evation data to derive harmonic constituents (Z0, M2, S2, N2, K1, M4, O1, M6, MK3, S4, MN4, nu2, S6, mu2, 2N2, OO1,
lambda2, S1, M1, J1, Mm, Ssa, Sa, MSF, Mf, rho1, Q1, T2, R2, 2Q1, P1, 2SM2, M3, L2, 2MK3, K2, M8, MS4) using pytides
0.0.4 (https://github.com/sam-cox/pytides). Third, we calculated storm surge anomalies by subtracting the astronomical tides
from the observed sea-surface elevation data. To account for annual variations, we averaged the anomalies at the time of each
typhoon's approach and subtracted this average. Finally, we selected 16 typhoons with maximum surge anomalies of 30 cm or
more at the Hakata tidal station.

### 2.3 Wind and air pressure

We obtained wind speed, wind direction, and air pressure at the sea surface for the period 1998 to 2022 from the Fukuoka
weather station (see Figure 1) which is the nearest station to the Hakata tidal station, provided by JMA (https://www.data.jma.
go.jp/stats/etrn/index.php)

### 2.4 Numerical simulation

An unstructured grid Finite-Volume Community Ocean Model (FVCOM, Chen et al. (2003)) was used to calculate the storm
surge associated with the input meteorological data for air pressure and winds. FVCOM utilizes the finite-volume method
and employs a triangular grid system in the horizontal and generalized terrain-following coordinates in the vertical (Yoon
and Shim (2013)). The model incorporates modified level 2.5 Mellor and Yamada (1982) and Smagorinsky (1963) turbulent
closure schemes for vertical and horizontal mixing. The governing equations include the equations of motion and the continuity
equation:





$$\frac{\partial u}{\partial t} + u\frac{\partial u}{\partial x} + v\frac{\partial u}{\partial y} + w\frac{\partial u}{\partial z} - fv = -\frac{1}{\rho_0}\frac{\partial(p_H + p_a)}{\partial x} + \frac{\partial}{\partial z}\left(K_m\frac{\partial u}{\partial z}\right) + F_u, \tag{1}$$

$$\frac{\partial v}{\partial t} + u\frac{\partial v}{\partial x} + v\frac{\partial v}{\partial y} + w\frac{\partial v}{\partial z} + fu = -\frac{1}{\rho_0}\frac{\partial(p_H + p_a)}{\partial y} + \frac{\partial}{\partial z}\left(K_m\frac{\partial v}{\partial z}\right) + F_v, \tag{2}$$

$$\frac{\partial w}{\partial t} + u\frac{\partial w}{\partial x} + v\frac{\partial w}{\partial y} + w\frac{\partial w}{\partial z} = \frac{\partial}{\partial z}\left(K_m\frac{\partial w}{\partial z}\right) + F_w, \tag{3}$$

$$\frac{\partial u}{\partial x} + \frac{\partial v}{\partial y} + \frac{\partial w}{\partial z} = 0, \tag{4}$$

where $x$, $y$ and $z$ are horizontal and vertical positions; $u$, $v$, and $w$ are the velocities in the $x$, $y$ and $z$ directions, respectively; $t$ is time; $p_a$ is air pressure at the sea surface; $p_H$ is hydrostatic pressure; $f$ is the Coriolis parameter; $g$ is the acceleration due to gravity; $K_m$ is vertical eddy viscosity; and $F_u$, $F_v$ and $F_w$ are the horizontal and vertical momentum diffusion terms, respectively.

The boundary conditions at the sea surface are as follows:

$$K_m\left(\frac{\partial u}{\partial z}, \frac{\partial v}{\partial z}\right) = \frac{1}{\rho_0}(\tau_{sx}, \tau_{sy}), \tag{5}$$

$$w = \frac{\partial \eta}{\partial t} + u\frac{\partial \eta}{\partial x} + v\frac{\partial \eta}{\partial y} \tag{6}$$

where $\eta$ is the height of the free surface. $\rho_0$ is a density of seawater. $\tau_{sx}$ and $\tau_{sy}$ are the $x$ and $y$ components of the sea surface shear stress and are expressed using the following equations:

$$(\tau_{sx}, \tau_{sy}) = \rho_a C_s U(U_{10}, V_{10}), \tag{7}$$

$$U = \sqrt{U_{10}^2 + V_{10}^2} \tag{8}$$

where $U_{10}$ and $V_{10}$ are the $x$ and $y$ components of the wind speed at 10 m above the sea surface, $\rho_a$ is a density of the atmosphere, and $C_s$ is the wind drag coefficient, which was calculated by Large and Pond (1981)'s formula:

$$C_s = \begin{cases} 1.20 \times 10^{-3} & \text{if} \quad 0 \leq U < 11\text{m/s} \\ (0.49 + 0.065U) \times 10^{-3} & \text{if} \quad 11\text{m/s} \leq U \end{cases} \tag{9}$$

The boundary conditions at the seafloor are as follows:

$$K_m\left(\frac{\partial u}{\partial z}, \frac{\partial v}{\partial z}\right) = \frac{1}{\rho_0}(\tau_{bx}, \tau_{by}), \tag{10}$$

$$w = -u\frac{\partial h}{\partial x} - v\frac{\partial h}{\partial y} \tag{11}$$

where $h$ is the water depth. $\tau_{bx}$ and $\tau_{by}$ are the $x$ and $y$ components of the seafloor shear stress and are expressed as follows:

$$(\tau_{bx}, \tau_{by}) = \rho_0 C_b \sqrt{u^2 + v^2}(u, v). \tag{12}$$



The bottom drag coefficient was determined by matching a logarithmic bottom layer to the model at a height $z_a$ above the bottom.

$$C_b = \max \left( \kappa^2 / \ln \left( \frac{z_a}{z_0} \right)^2, 0.0025 \right) \tag{13}$$

where $k = 0.4$ is the von Karman constant, and $z_0$ is the bottom roughness parameter, which is basically 0.001 m in the ocean.

The model domain is shown in Figure 2. It spans an extensive area from 125° to 135° E in longitude and 27° to 36° N in latitude (Figure 2a). Coastal lines are sourced from the Ministry of Land, Infrastructure, Transport and Tourism in Japan (https://nlftp.mlit.go.jp/ksj/gml/datalist/KsjTmplt-C23.html). Water depth data are obtained from the Japan Hydrographic Association (https://www.jha.or.jp/jp/shop/products/btdd/) and the Japan Oceanographic Data Center (https://www.jodc.go.jp/jodcweb/JDOSS/infoJEGG_j.html) and are interpolated to the mesh using inverse distance weighting. A large unstructured grid with a resolution of 50 km was used for the open sea, while very small triangular meshes with a resolution of 300 m were employed along Kyushu Island, including Hakata Bay. The model uses three sigma levels in the vertical. The triangular grid consists of 254,335 nodes and 483,582 elements. Winds at a height of 10 m above the sea surface and air pressure at the sea surface are obtained from MSM (https://www.data.jma.go.jp/suishin/cgi-bin/catalogue/make_product_page.cgi?id=MesModel), which provides grid point values from an hourly weather forecast for Japan and its surrounding ocean area, calculated at a finer grid spacing (5 km) than the global model. For each typhoon, the computation period is 8 days with a time step of 1.0 s. The initial water level is set at mean sea level, and tides and wind waves are not considered.



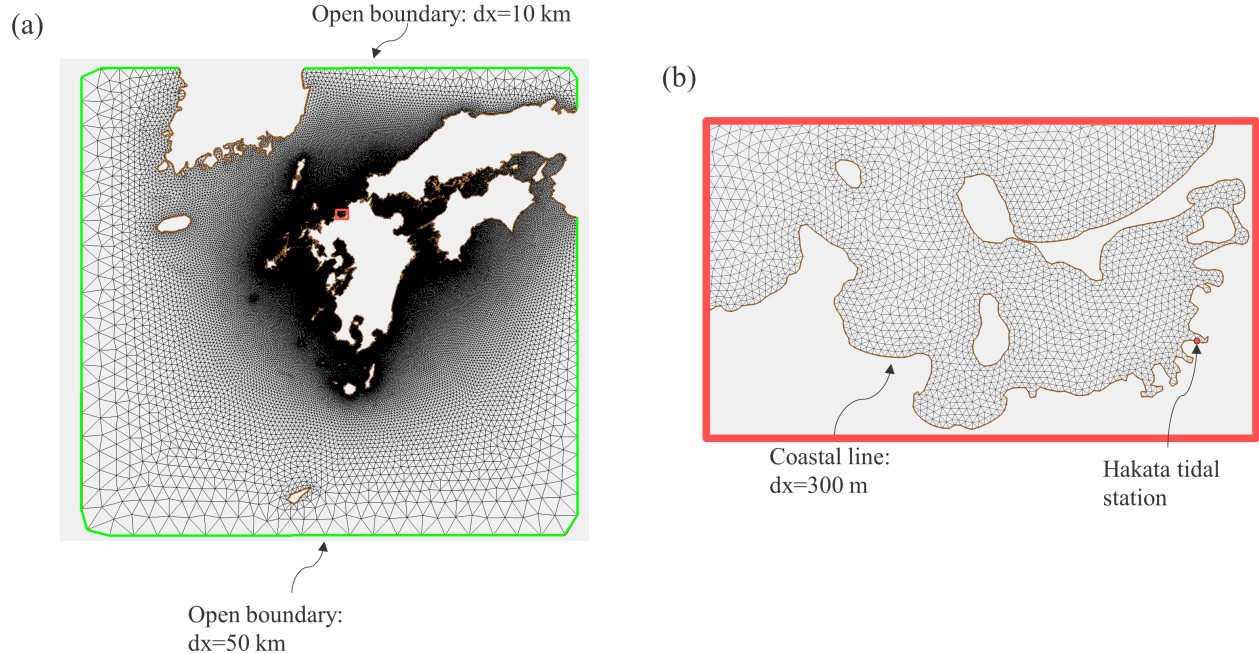

**Figure 2.** Overview of the model domain ((a) grid composition; (b) enlarged grid view of Hakata Bay).





## 2.5 Wavelet transform

We conducted spectral analysis using continuous wavelet transform (CWT) to investigate the oscillatory components within the storm surges. CWT is a method suitable for non-stationary spectral analysis. In CWT, a small wave known as the mother wavelet is utilized. Frequency analysis is performed by comparing the mother wavelet with the wave of interest. The mother wavelet is given by:

$$\psi_{a,b}(t) = \frac{1}{\sqrt{a}}\psi\left(\frac{t-b}{a}\right) \tag{14}$$

where, $\psi(t)$ is the mother wavelet, $t$ is time, $a$ is a scale and $b$ is shift. When $a$ increases, the mother wavelet is stretched, making it more suitable for analyzing longer waves. Increasing $b$ results in a parallel shift in the positive direction along the time axis. For the mother wavelet defined by scale $a$ and shift $b$, denoted by $\psi_{a,b}(t)$, the continuous wavelet transform is expressed as follows:

$$W_{\psi_{a,b}}[x(t)] = \int_{-\infty}^{\infty} x(t)\psi_{a,b}^{*}(t)dt \tag{15}$$

where $x(t)$ is the signal of interest and $*$ denotes the complex conjugate. By continuously varying the scale $a$ and shift $b$, the continuous wavelet transform maps to a two-dimensional plane of $(a,b)$, allowing for frequency analysis in the time domain. In this study, the following complex Morlet wavelet is utilized as the mother wavelet:

$$\psi(t) = \frac{1}{\sqrt{\pi B}}\exp\left(-\frac{t^2}{B}\right)\exp(2\pi C j t) \tag{16}$$

where $j$ is the imaginary unit, $B = 1.5$ is the bandwidth, and $C = 1.0$ is the center frequency. The values are recommended
and lead to good a resolution or artifact (Lee et al. (2019)).

## 3 Result and discussion

### 3.1 Time series analysis

In this section, we examine the observed data such as storm surge anomalies, wind, and atmospheric pressure, and categorize them with the typhoon's tracks.

Figure 3 illustrates the tracks of typhoons that passed within a 300 km radius of the Hakata tidal station from 1999 to 2022. The typhoon paths can be categorized into two patterns. The first, WCTS-type typhoons follow a route from the SCKP to WCTS. This category includes typhoons such as Rusa (2002), Maemi (2003), Megi (2004), Sanba (2012), Chaba (2016), Kong-rey (2018), Maysak (2020), Haishen (2020), and Hinnamnor (2022). The second, ECTS-type typhoons travel from the ECTS to the NCKI, including Bart (1999), Chaba (2004), Songda (2004), Shanshan (2006), Danas (2013), Goni (2015), and
Tapah (2019).





Figure 4 displays the time series of storm surge anomalies at the Hakata tidal station for both WCTS-type and ECTS-type typhoons. For Bart (1999), no calculated values are available due to the absence of meteorological data (MSM). For both WCTS-type and ECTS-type typhoons, the calculated values exhibit a pronounced 2-h periodic seiche at Hakata Bay(Yamashiro et al. (2016)), but the general trend of the time series is well represented.

ECTS-type's anomalies tend to rise sharply over a short period. Chaba (2004), Bert (1999), and Tapha (2019) each had a single peak in maximum storm surge anomaly, while Songda (2004), Shanshan (2006), and Goni (2015) experienced a second peak within approximately two hours before or after the time of the maximum anomaly. Additionally, the maximum storm surge anomaly occurs when the typhoon approaches NCKI. As a representative of the ECTS-type, Figure 5a shows the time series of wind and air pressure at the Fukuoka weather station during Goni (2015). At 8:00 on August 25, 2015, when the storm

surge anomaly was at its maximum (Figure 4a: purple arrow), the air pressure dropped to 970 hPa, and a northerly wind was blowing (Figure 5a).

Therefore, the rapid rise in storm surge anomaly for ECTS-type typhoons can be explained by the drop in air pressure due to the typhoon's approach, causing the sea level to rise through the inverted barometer effect, and the northerly wind pushing seawater towards the land, further raising the sea level through wind set-up. Moreover, the appearance of a second peak during

the passage of Songda (2004), Shanshan (2006), and Goni (2015) can be attributed to the fact that the peak of the static water level rise inverted barometer effect and wind set-up did not coincide with the peak of the 2-hour bay water oscillation in Hakata Bay (Yamashiro et al. (2016)).

On the other hand, the storm surge anomaly of WCTS-type typhoons tends to show a rapid decrease after the first peak, followed by a second peak where the maximum value occurs (Figure 4b: purple arrow). Additionally, the first peak occurred

when the typhoon approached Hakata Bay (Figure 3: green dot). The second peak, except in the case of Sanba (2012), occurred when the typhoon entered the Sea of Japan (Figure 3: blue dot). Figure 5b shows the time series of wind and pressure at the Fukuoka weather station during Hinnamnor (2022). The first surge peak coincided with the typhoon's closest approach to NCKI, when the minimum pressure was recorded at 991.4 hPa, thus explaining the first peak through the inverted barometer effect (Figure 4b and Figure 5b: red arrow). However, the second peak occurred when the typhoon had already entered the

Sea of Japan, at which time the pressure at NCKI had risen to 1,000 hPa, and the wind speed was below 5 m/s (Figure 4b and Figure 5b: purple arrow). Thus, the second peak cannot be explained by the inverted barometer effect or wind set-up. Therefore, we aim to elucidate the mechanism behind the occurrence of the second peak surge.





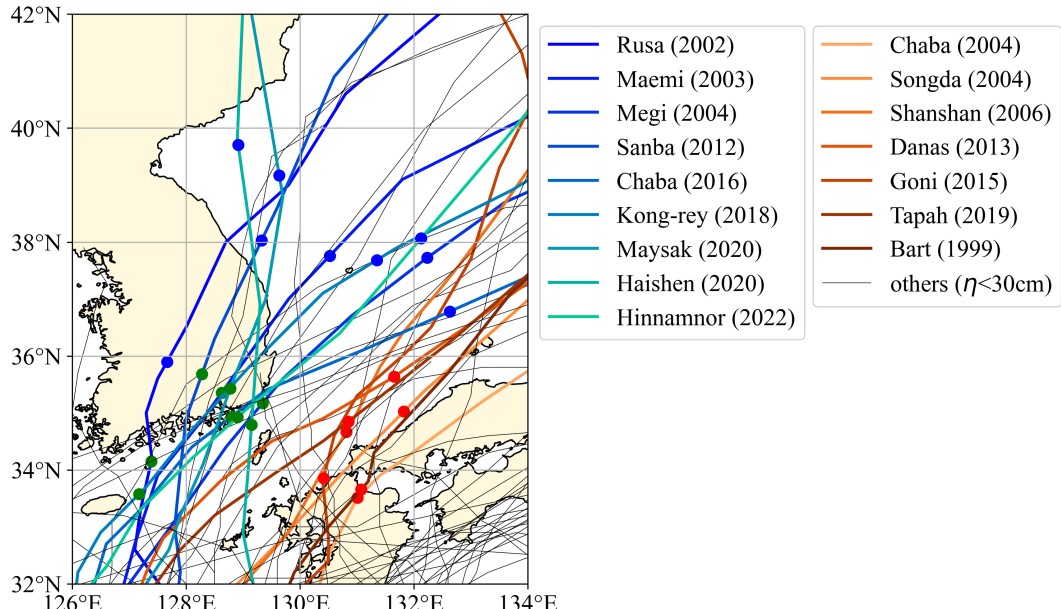

**Figure 3.** Tracks of typhoons passing within a 300 km radius of the Hakata Bay tide station from 1999 to 2022. The red or blue colored lines indicate the 16 typhoon paths that caused storm surge anomalies of 30 cm or more in the Hakata Bay tidal station, and the black lines indicate the remaining typhoons. The red colored lines indicate the paths that passed from ECTS to over NCKI (ECTS-type), and the blue colored lines indicate the paths that passed from the SCKP to WCTS (WCTS-type). For WCTS-type, green points refer to the time when first peak occurred at Hakata tidal station and blue points refer to the time when second peak occurred there. For ECTS-type, red points refer to the time when the anomaly is maximum (see arrows in Figure 4).





**Figure 4.** The timeseries of storm surge anomalies at the Hakata tidal station ((a) ECTS-type; (b) WCTS-type). The black line shows the observed data, and the red line is the simulated anomaly. The red, blue, and purple arrows drawn for the observed data indicate the time at which the first peak, local minimum, the second peak or maximum is observed, respectively.



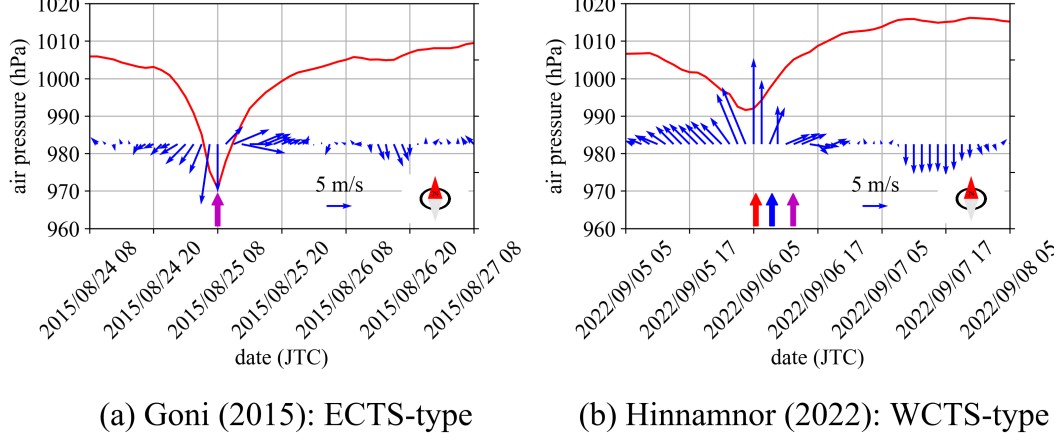

(a) Goni (2015): ECTS-type  (b) Hinnamnor (2022): WCTS-type

**Figure 5.** The time series of air pressure and winds at Fukuoka weather station. (a) ECTS-type; (b) WCTS-type.



## 3.2 Spacial distribution analysis

In this section, we clarify the cause and characteristics of the second peak shown in WCTS-type by conducting a detailed
analysis of the spatial distribution of storm surge anomalies, wind, and atmospheric pressure during Hinnamnor (2022), as
representative of the WCTS-type.

Figure 6 illustrates the spatial distribution of the storm surge anomaly during Hinnamnor (2022). Figure 7 shows the spatial
distribution of atmospheric pressure and wind. The upper panels of Figure 8 show the time series of the calculated tidal levels
at the Hakata Bay tidal station when the external forces of pressure and wind are independently applied.

At any given time, the storm surge anomaly in panel (a) of Figure 6 equals the sum of the anomalies shown in panels
(b) and (c) of Figure 6. In other words, for the WCTS-type, the storm surge anomaly caused by atmospheric pressure and
winds is independent of each other. Therefore, by investigating the development processes of the storm surge anomaly due to
atmospheric pressure and wind separately, we can consider the storm surge development mechanisms when both pressure and
wind are applied simultaneously.

At 2:00 on September 6, 2022, the typhoon enters the Tsushima Strait. At this time, the sea level rises in accordance with
the low air pressure distribution around the typhoon's center (Figure 6b0 and Figure 7a0), resulting in a sea level rise of
approximately 20 cm at Hakata Bay located at the center of NCKI (Figure 8a: upper panel). On the other hand, the storm surge
induced by the winds gradually increases from NCKI towards SCKP, forming a gradient (Figure 6c0). This is because the wind
is blowing towards SCKP, causing a storm surge rise due to wind set-up (Figure 7b0).

At 5:00 on September 6, 2022, the typhoon was near the SCKP, making its closest approach to the NCKI. At this time, with
only atmospheric pressure applied, a positive anomaly develops around the typhoon, leading to a positive anomaly throughout
the Tsushima Strait (Figure 6b1). The distribution of positive anomalies corresponds with the distribution of the low-pressure
system (Figure 7a1). As a result, the storm surge at Hakata Bay rises by approximately 20 cm due to inverted barometer effect
(Figure 8a: upper panel).

When wind is applied, positive anomalies appear along the SCKP and the NCKI, although these values are relatively smaller
than those at 5:00 on September 6, 2022 (Figure 6c1). The reason for this is that the wind blowing towards SCKP abruptly
shifted to a westerly wind as the typhoon moves northeast, disrupting the balance and causing the gradient of storm surge to
begin to collapse (Figure 7b1).

By 8:00 on September 6, 2022, as the typhoon entered the Sea of Japan, a negative anomaly developed in the Tsushima
Strait (Figure 6a2), which is mainly derived from wind induced anomaly (Figure 6c2). By 11:00 on September 6, 2022, as the
typhoon continued northeastward in the Sea of Japan, a positive anomaly appeared in the NCKI (Figure 6a3), which is mainly
derived from the positive anomaly induced by wind (Figure 6c3).

By 14:00 on September 6, 2022, this positive anomaly in the NCKI intensified, resulting in the second peak (Figure 6a),
which is mainly derived from positive anomalies induced by both air pressure and wind (Figure 6b4, c4). The atmospheric
pressure across the Tsushima Strait had risen to near normal levels, and wind speeds were below 10 m/s after 8:00 on September
6, 2022 (Figure 7a2-4, b2-4). Consequently, the only forces acting on the seawater in the Tsushima Strait are gravity and the



Coriolis force, with no external forces to forcibly raise the storm surge. This second peak arises from the superposition of oscillations generated in the Tsushima Strait by both atmospheric pressure and wind, with the second peak occurring in the NCKI when the peaks of these oscillations coincide.



(a) Air pressure and wind induced anomaly.    (b) Air pressure induced anomaly.    (c) Wind induced anomaly.

**Figure 6.** The distribution of storm surge anomaly during Hinnamnor (2022). The black line shows the path of the typhoon, and the black star indicates the position of the center of the typhoon. (a) represents the storm surge anomaly when both air pressure and wind are applied as external forces, (b) represents the case where only air pressure is applied, and (c) represents the case where only wind is applied.



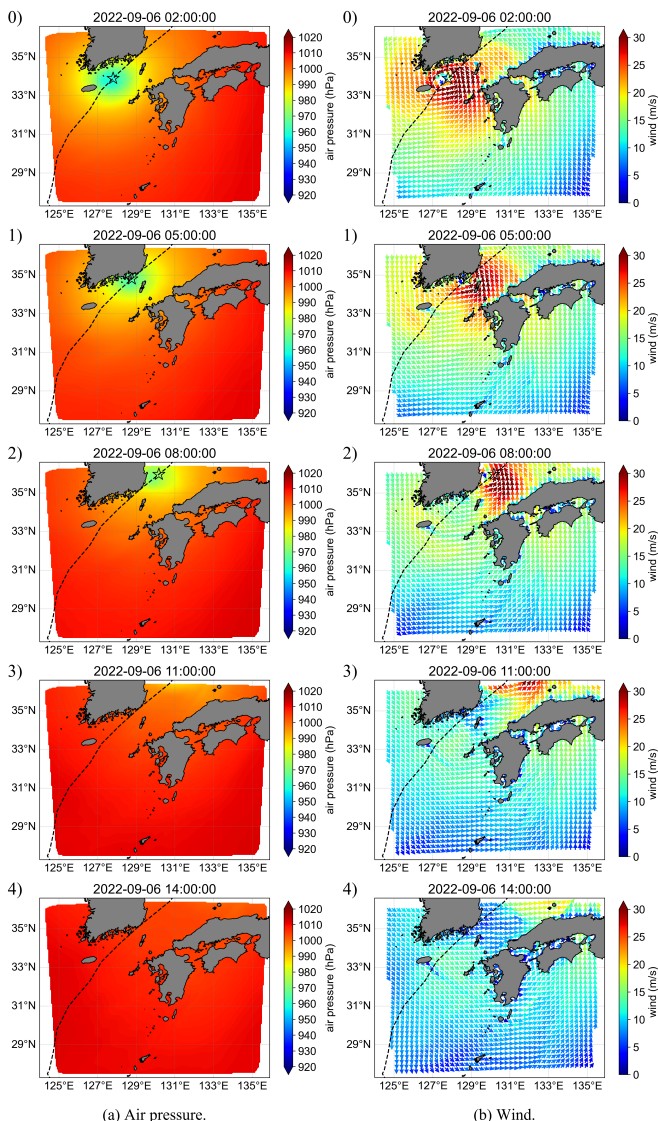

(a) Air pressure.          (b) Wind.

**Figure 7.** The distribution of air pressure and wind during Hinnamnor (2022). The black line shows the path of the typhoon, and the black star indicates the position of the center of the typhoon. (a) represents the distribution of air pressure, and (b) represents the distribution of wind.





### 3.3 Mchanism of storm surge dvelopment

#### 3.3.1 Wavelet transform

This chapter discusses the factors contributing to the second peak during the passage of a WCTS-type typhoon. As shown in Figure 4, the storm surge anomaly fluctuates due to the superposition of oscillations with multiple periods. To identify the oscillatory components other than tidal constituents present in the storm surge anomaly time series in Hakata Bay, continuous wavelet transform (CWT) is applied.

The lower panels of Figure 8 present the results of the continuous wavelet transform, referred to as scalograms, where atmospheric pressure (a) and wind (b) are applied independently as external forces. In both (a) and (b), three types of oscillations with periods of 10 h, 5 h, and 2 h are observed.

Figure 9 shows the timeseries of oscillatory components with periods of 2 h, 5 h, and 10 h in Hakata Bay by inversed CWT. The 2-h period oscillation is a harbor oscillation at Hakata Bay (Yamashiro et al. (2016)). This oscillation occurred both before the typhoon approached Hakata Bay and after it passed over (Figure 9a). Meanwhile, when atmospheric pressure is applied as the external force, the 5-h period oscillation reaches its maximum amplitude at 13:00 on September 6, 2022 (Figure 9b: cal_p). When wind is the external force, this maximum amplitude occurs 2 h earlier, at 11:00 on September 6, 2022 (Figure 9b: cal_w). In both cases, when atmospheric pressure and wind are the external forces, the 5-h period oscillation coincides with the second peak in the storm surge anomaly, following a minimum. Similarly, for the 10-h period oscillation, the peak occurs about 2 h earlier when wind is the external force (Figure 9c). The maximum amplitude is reached between 11:00 and 17:00 on September 6, 2022, corresponding to the second peak in the storm surge anomaly, with the oscillation continuing afterward at a decreasing amplitude. Figure 10 and Figure 11 show the spatial distribution of the 5-h and 10-h period oscillatory components in the Tsushima Strait, respectively. For the 5-h period oscillation, regardless of the external force, Tsushima Island, the boundary between the East China Sea and the Tsushima Strait, and the boundary between the Sea of Japan and the Tsushima Strait act as nodes, while the SCKP and the NCKI serve as antinodes. Additionally, the SCKP and NCKI are in opposite phases. The timing of the oscillation differs depending on the external force: when atmospheric pressure is the external force (Figure 10a), the oscillation begins at 7:00 on September 6, 2022, when the typhoon is located on the eastern coast of Korea; when wind is the external force (Figure 10b), the oscillation starts at 5:00 on September 6, 2022, as the typhoon is located near the SCKP, with wind forcing occurring 2 h earlier. For the 10-h period oscillation, the boundary between the East China Sea and the Tsushima Strait, and the boundary between the Sea of Japan act as nodes, while Tsushima Island, the SCKP, and the NCKI serve as antinodes. Unlike the 5-h period oscillation, the SCKP and NCKI are in the same phase. The timing of the 10-h period oscillation is similar to that of the 5-h period oscillation, with wind forcing occurring 2 h earlier.

Considering that both atmospheric pressure and wind had returned to normal conditions by the time the 5-h and 10-h period oscillations occurred at the NCKI, these oscillations are likely driven by the natural frequencies induced by the topography seiche rather than by external forces. It can be inferred that the second peak in the storm surge anomaly is a result of the superposition of these two oscillatory components.





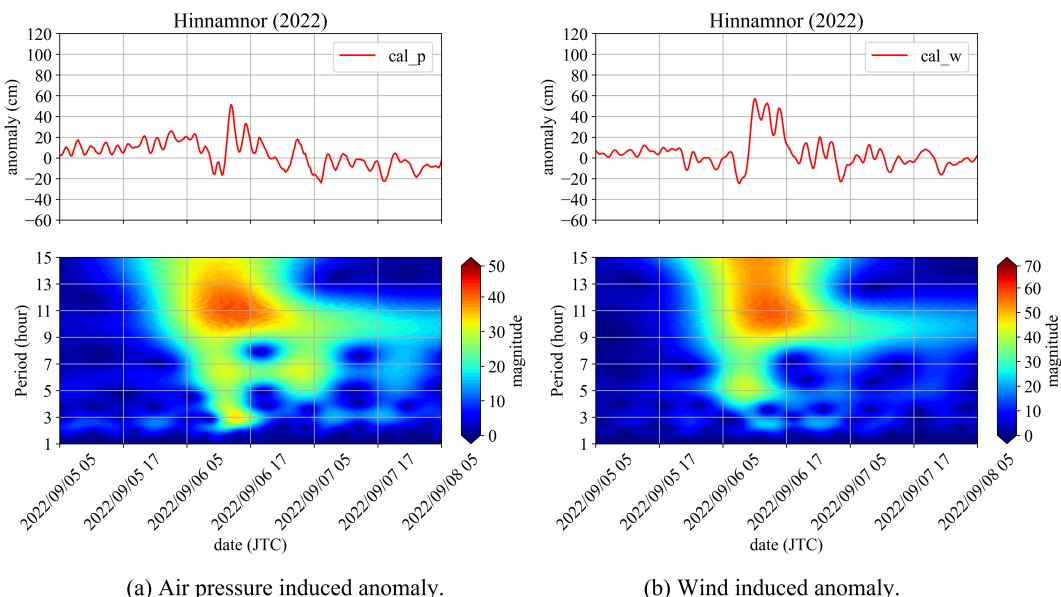

(a) Air pressure induced anomaly.      (b) Wind induced anomaly.

**Figure 8.** Simulated time series and scalograms at the Hakata tidal station. (a) shows the case where the anomaly induced by air pressure is transformed, and (b) shows the case where the anomaly induced by wind is transformed.



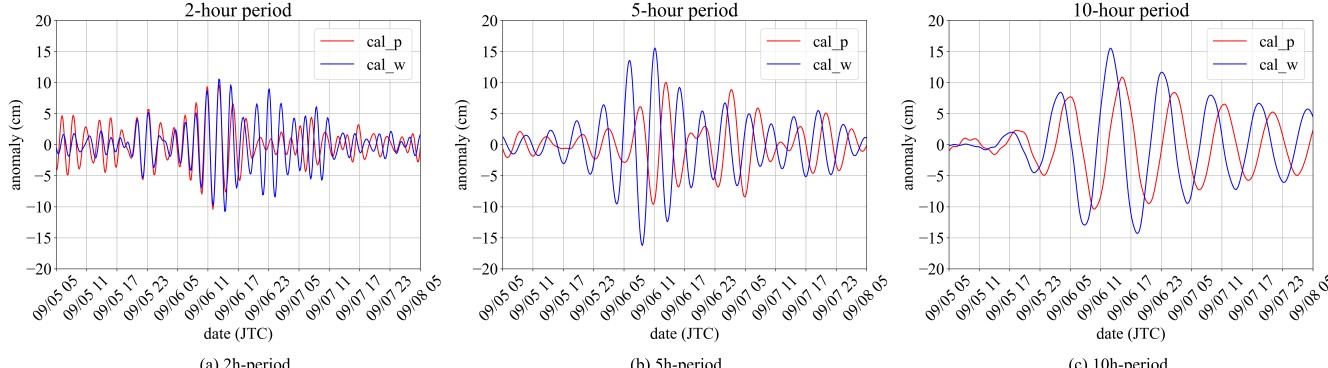

**Figure 9.** Timeseries of oscillatory components with periods of (a) 2 hours, (b) 5 h, and (c) 10 h at the Hakata tidal station by inversed CWT. The red and blue lines show air pressure and wind induced anomalies, respectively.





(a) Air pressure induced anomaly.     (b) Wind induced anomaly.

**Figure 10.** The distribution of 5-h period oscillations in Tsushima Strait.





(a) Air pressure induced anomaly.  (b) Wind induced anomaly.

**Figure 11.** The distribution of 10-h period oscillations in Tsushima Strait





### 3.3.2 Development of storm surge anomalies by oscillation modes using wave equation's analytical solution.

The reason both the 5-h and 10-h period oscillations form nodes at the boundaries between the Tsushima Strait and the Sea of
Japan, as well as between the Tsushima Strait and the East China Sea, is due to the rapid increase in water depth. As shown
in Figure 1, there is a significant change in water depth at these boundaries. When such a discontinuous change in water depth
occurs, it is known that the amplitude of oscillations decreases exponentially near the discontinuity. For example, the waveform
of edge waves when water depth changes discontinuously is described by the following equation.

$$\eta_1 = a\cos(\mu_1 x)\cos(ky + \sigma t) \tag{17}$$

$$\eta_2 = a\cos(\mu_1 l)e^{-\mu_2(x-l)}\cos(ky + \sigma t) \tag{18}$$

$$\mu_1 = \sqrt{\frac{\sigma^2}{gh_1} - k^2}, \quad \mu_2 = \sqrt{k^2 - \frac{\sigma^2}{gh_2}} \tag{19}$$

Let the coastline lie along the $y$-axis, with $x > 0$ representing the sea. The water depth and width of the continental shelf are
denoted as $h_1$ and $l$, respectively, while the water depth of the deep sea is $h_2$, with $h_1 < h_2$ (i.e., the depth of the deep sea is
greater than that of the continental shelf). The wave-number and angular frequency are represented by $k$ and $\sigma$, respectively,
and $a > 0$ is a constant. According to Equation (18), the amplitude of the oscillation in the region where $x > l$ (the deeper area)
becomes significantly smaller compared to the region where $x < l$ due to the discontinuous change in water depth.

This phenomenon arises from the properties of long waves. The group velocity of long waves is expressed as $C = \sqrt{gh}$,
meaning that waves move slower in shallow regions and faster in deeper regions. As a result, when long waves generated in
shallow areas attempt to enter deeper regions, their group velocity increases rapidly. This sudden shift in velocity disrupts the
wave's penetration into the deeper water, causing part of the incoming wave to reflect as if encountering a fixed boundary at the
point where the water depth changes. The interaction between the reflected and incoming waves forms standing waves, creating
a fluid oscillatory system where specific oscillation periods dominate. Consequently, in much deeper regions, oscillations do
not occur, leading to the formation of nodes. This explains why the 5-h and 10-h period oscillations are confined to the shallow
regions, where these standing wave patterns develop.

Here, we examine the reasons behind the 5-h period oscillation creating a node in the region between the SCKP and the
NCKI, near Tsushima Island. This phenomenon is related to the oscillation without Coriolis force. By solving the wave equation:

$$\frac{\partial^2 \eta}{\partial t^2} = c^2\left(\frac{\partial^2 \eta}{\partial x^2} + \frac{\partial^2 \eta}{\partial y^2}\right) \tag{20}$$

$$c = \sqrt{gh} \tag{21}$$

in a rectangular domain with dimensions $b \times l$, where the depth $h$ is constant, and ensuring the solution satisfies the boundary
conditions:

$$v = 0, \quad (y = 0, b) \tag{22}$$

$$\eta = 0, \quad (x = 0, l) \tag{23}$$



the solution is obtained:

$$\eta = A\sin(kx)\cos(k'y)\sin(\sigma t) \tag{24}$$

$$k = \frac{m}{l}\pi, \quad k' = \frac{n}{b}\pi, \quad \sigma = c\sqrt{k^2 + k'^2} \tag{25}$$

Here, $m$ and $n$ are integers representing the modes of vibration in the $x$-axis and $y$-axis directions, respectively.

Additionally, the period of oscillation is given by

$$T_{m,n} = \frac{2\pi}{\sigma} = \frac{2}{\sqrt{gh}}\left\{\left(\frac{m}{l}\right)^2 + \left(\frac{n}{b}\right)^2\right\}^{-1/2} \tag{26}$$

For the spatial scale of the Tsushima Strait, with $h = 80$ m, $l = 550$ km, $b = 250$ km and $A = 5$ cm, the solutions for $(m,n) = (1,1),(1,0)$ are presented in Equation (24) and depicted in Figure 12.

The oscillation with $(m,n) = (1,1)$ has nodes at $x$=0 and $l$, $y=\frac{b}{2}$, and antinodes at $x=\frac{l}{2}$, $y$=0 and $b$. The oscillation with $(m,n) = (1,1)$ has nodes at $x$=0 and $l$, $y=\frac{b}{2}$, and antinodes at $x=\frac{l}{2}$, $y$=0 and $b$. The period of this oscillation is $T_{1,1}$=4.52 h. Additionally, the oscillation with $(m,n) = (1,0)$ has nodes at $x$=0 and $l$, and antinodes at $x=\frac{l}{2}$. The period of this oscillation

is $T_{1,0}$=10.91 h. These periods correspond to the 5-h and 10-h periods observed in the CWT and shows that these oscillations that cause delayed storm surges if the presence of the Coriolis force is ignored.

Therefore, during the passage of WCTS-type typhoons, there are two peaks in storm surge at the NCKI (Figure 4b: red arrow and purple arrow). The first peak can be attributed to the barometer effect (Figure 6b1 and 7a1). As the typhoon progresses into the Sea of Japan, the constraints from wind set-up and the barometer effect are alleviated, releasing potential energy and

resulting in oscillations with periods of approximately 5 h in the $(1,1)$ mode and 10 h in the $(1,0)$ mode (Figure 10 and 11). The timing of oscillation generation differs between external forces such as atmospheric pressure and wind due to the variation in constraint removal timing. For atmospheric pressure, the peak water level at the SCKP occurs when the typhoon is directly overhead on SCKP (Figure 6b1 and 7a1). In contrast, for wind, the peak water level occurs when the prevailing southeast wind at the SCKP causes the maximum onshore push, which happens earlier (Figure 6c0 and 7b0). By the time the peak water

level due to atmospheric pressure is reached, the constraint from wind has already been released, leading to earlier onset of oscillations due to wind. That is why the second peak in storm surge at the NCKI results from the superposition of these two oscillation modes.

In Typhoon Hinnamnor (2022), the second peak is well-explained by the superposition of the $(1,1)$ and $(1,0)$ mode oscillations. However, in cases such as Maysak (2020) and Haishen (2020), where high tidal deviations persisted long after the second

peak (Figure 4), it was observed that the southwesterly winds following these typhoons caused Ekman transport, pushing seawater toward the NCKI (Niimi et al. (2022)). This results in a situation where the water level remains elevated even after the second peak. Prolonged high storm surge increases the probability of occurring high astronomical tides and high storm surge at the same time, raising the risk of flooding. Typhoons that maintain high water levels after the second peak are often those moving north through the Tsushima Strait within the WCTS-type category (Figure 3). It is believed that even minor differences

in their paths can lead to variations in the persistence of storm surge deviations. Future research will focus on identifying




WCTS-type typhoons that generate the most dangerous storm surges by considering Ekman transport, oscillations in the $(1,1)$ and $(1,0)$ modes, and the timing of the release of external forces.

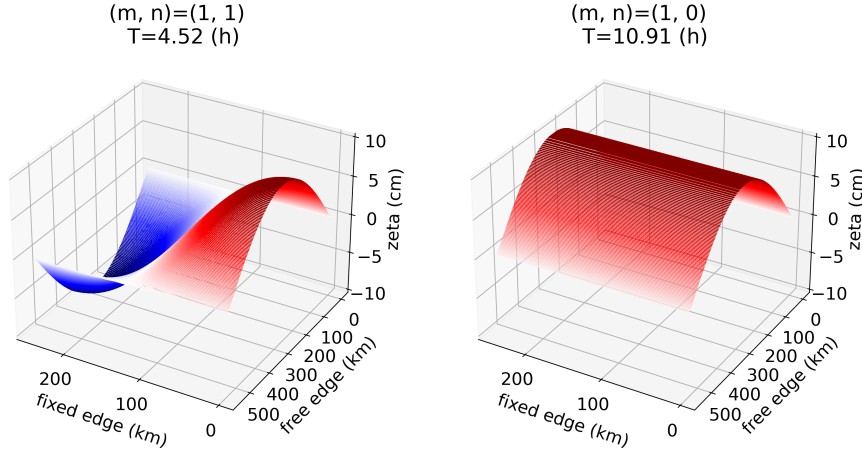

**Figure 12.** Solutions of the wave equation. $m$ and $n$ represent the mode of vibration in the free edge direction and fixed edge direction, respectively. Color corresponds to zeta.



## 4 Conclusions

This study investigated the mechanisms of storm surge development at NCKI using observational data, storm surge numerical
simulations, and continuous wavelet transforms. Typhoons that cause significant storm surges at NCKI can be classified based
on their paths: one type passes from ECTS to NCKI, and the other passes from the SCKP through WCTS.

For ECTS-type typhoons, when the storm is closest to NCKI, the pressure drops to its lowest, and north winds push seawater
towards NCKI, causing the storm surge peak. Therefore, the storm surge development process for ECTS-type typhoons can
be explained by the barometer effect and wind set-up. In contrast, WCTS-type typhoons cause an initial peak in storm surge
at the time of the closest approach, followed by a rapid decrease. However, a second peak in storm surge, which is larger than
the first, occurs later, resulting in a significant storm surge at NCKI after the typhoon has passed. By the time of the second
peak, atmospheric pressure has returned to near-normal levels and wind speeds are low, rendering the barometer effect and
wind set-up insufficient to explain this phenomenon.

Spectral analysis using continuous wavelet transforms revealed a continuous 2-h periodic oscillation, with oscillations of
5-h and 10-h periods emerging after the first peak. The 2-h period oscillation is localized to Hakata Bay. To investigate the
characteristics of the 5-h and 10-h period oscillations, storm surge numerical simulations were performed, and spectral analysis
was conducted on the obtained spatial distributions using continuous wavelet transforms. The 5-h period oscillation, charac-
terized by antinodes at NCKI and the SCKP and nodes at Tsushima Island, represents the $(1,1)$ mode of the Tsushima Strait's
natural oscillation. The 10-h period oscillation corresponds to the $(1,0)$ mode of the Tsushima Strait's natural oscillation.
These oscillations arise from the release of potential energy in the Tsushima Strait when the typhoon enters the Japan Sea. The
superposition of these two mode oscillations leads to the delayed second peak in storm surge at NCKI.

We identified the occurrence of anomalous storm surges caused by typhoons passing through a strait under specific conditions
and conducted a detailed investigation of their generation mechanisms. These mechanisms are not limited to the studied region;
they can occur in any area where typhoons or low-pressure systems pass through a strait. However, it is important to note that
the oscillation period, amplitude, and the components contributing to the oscillations vary, leading to differences in the timing
and magnitude of storm surges.

*Code and data availability.* The datasets and files generated during and/or analyzed during the current study are available from the corre-
sponding author upon reasonable request.

*Author contributions.* Conceptualization, Shinichiro Ozaki, Yoshihiko Ide and Masaru Yamashiro; Data curation, Shinichiro Ozaki; For-
mal analysis, Shinichiro Ozaki; Funding acquisition, Shinichiro Ozaki; Investigation, Shinichiro Ozaki and Yoshihiko Ide; Methodology,
Shinichiro Ozaki and Yoshihiko Ide; Project administration, Masaru Yamashiro; Resources, Shinichiro Ozaki; Software, Shinichiro Ozaki
and Yoshihiko Ide; Supervision, Yoshihiko Ide and Masaru Yamashiro; Validation, Shinichiro Ozaki; Visualization, Shinichiro Ozaki; writing



– original draft preparation, Shinichiro Ozaki; Writing – review  editing, Shinichiro Ozaki, Yoshihiko Ide and Masaru Yamashiro. All authors have read and agreed to the published version of the manuscript.

*Competing interests.* The authors declare that they have no known competing financial interests or personal relationships that could have appeared to influence the work reported in this paper.

*Acknowledgements.* This work was supported by JST BOOST, Japan Grant Number JPMJBS2406.

**Declaration of generative AI and AI-assisted technologies in the writing process**

During the preparation of this work the authors used Chat GPT in order to improve language. After using this tool, the authors
reviewed and edited the content as needed and take full responsibility for the content of the publication.





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
