# Peer review of "Mechanism of Delayed Storm Surges in Straits: Seiche-Induced Oscillations Triggered by Typhoon Passage"

_EGUsphere, 2024_

## Author Response (AR2)

**Point-by-Point Response to Reviewer Comments and List of Changes**

We sincerely thank the reviewers for their insightful comments and constructive suggestions, which have significantly improved the quality and clarity of our manuscript. Below, we provide a detailed, point-by-point response to each comment and list all relevant changes made in the revised manuscript.

**Reviewer 1**

**Major Points**

**1. Governing Equations and Coordinate Consistency**

- Reviewer Comment: "The governing equations by Eqs. (1)-(4) use the Cartesian coordinate, but the numerical results use unstructured grids. It should be consistent with the model."
- Author's Response: Thank you for pointing this out. The governing equations are written in a Cartesian coordinate system, which remains valid when using unstructured grids in the finite volume framework. FVCOM utilizes the finite volume method for spatial discretization and can be applied using either Cartesian (xy) coordinates or geographic coordinates (latitude and longitude). For a detailed description of the discretization procedures, please refer to Chen et al. (2003).
- Changes in Manuscript: Clarification regarding the applicability of Cartesian coordinates with unstructured grids in FVCOM is implicitly addressed by the reference to Chen et al. (2003) and stated it in lines 174-175 of the revised manuscript. We further stated (lines 197-198) that for FVCOM input, xy coordinate data corresponding to latitude and longitude were used, and latitude and longitude were utilized for visualization.

**2. Quantitative Analysis of Oscillation Modes**

- Reviewer Comment: "The discussion about the oscillation modes between Korea and Kyushu
  in 3.3.2 is quite interesting and important. However, the analysis is qualitative and needs to
  improve more quantitatively."
- Author's Response: Thank you for this insightful comment. In the revised manuscript, we've significantly enhanced the quantitative and qualitative discussion of the oscillation modes. Specifically, in Section 6.2 "Dominant Oscillation Modes Identified by CWT", we've added a more quantitative analysis of the amplitudes associated with each oscillation mode. Furthermore, Section 7 "Discussion" now includes a comprehensive qualitative interpretation of these modes, along with comparisons to relevant literature to highlight the novelty of our findings.
- Changes in Manuscript: New quantitative analysis of oscillation mode amplitudes has been added to Section 6.2 "Dominant Oscillation Modes Identified by CWT". Section 7 "Discussion" has been expanded to include a detailed qualitative interpretation and comparisons with other literature.

**3. Improvement of English Writing**

- Reviewer Comment: "English writing needs to be improved before acceptance."
- Author's Response: We have carefully revised the manuscript for grammatical accuracy and clarity. The manuscript has been edited by a native English speaker to enhance the overall quality of the language.
- Changes in Manuscript: The entire manuscript has undergone comprehensive revision for grammatical accuracy, clarity, and overall English writing quality by a native English speaker.

**Minor Points**

**1. Expression of "s" as Storm Surge**

- Reviewer Comment: "A storm surge is a sea surface anomaly from the astronomical tide.
   Therefore, the expression of s is incorrect. It should be reworded as storm surge or storm surge height."
- Author's Response: We appreciate the reviewer's comment regarding the terminology of storm surge. Upon re-evaluating our usage, we have ensured that "storm surge" refers to the physical phenomenon of the abnormal sea level rise itself, caused by direct meteorological forcing (e.g., wind setup). When quantifying this rise, we have adopted "storm surge height." Regarding the use of "storm surges" (plural), while less common for the general phenomenon, it can be acceptable when referring to multiple, distinct storm surge events or types. Our manuscript has undergone native English speaker review, and this usage was deemed acceptable in context, aligning with some existing literature. Furthermore, to clearly differentiate between the general phenomenon and the specific deviation from astronomical tide, we have primarily used "storm surge anomaly" throughout the manuscript to represent the observed water level elevation attributable to meteorological effects, after removing the astronomical tide. This term specifically conveys the "abnormal sea level" aspect that the reviewer points out, effectively serving as our measure for the "water level rise amount" in our analysis. We believe these clarifications and adjustments enhance the precision of our terminology.
- Changes in Manuscript: We have refined the terminology related to water level variations. In the Introduction, "storm surge" and "storm surges" are now exclusively used when describing the general phenomenon of water level rise induced by wind and atmospheric pressure. For all other instances referring to the abnormal sea level, we consistently use "Storm Surge Anomaly".

**2. Organization of Sections and Paragraphs**

- Reviewer Comment: "There are many small subsections, mainly in section 2. There are also small paragraphs consisting of a few sentences. The size of sections, subsections, and paragraphs should be carefully considered following the standard manner of academic writing."
- Author's Response: We appreciate the suggestion regarding the structure of the manuscript. In the revised version, we have reorganized Section 2 by consolidating smaller subsections and expanding brief paragraphs where necessary.
- Changes in Manuscript: Section 2 (now parts of "Study Area," "Selected Typhoons," and
  "Observed Storm Surge and Weather Data") has been reorganized, consolidating smaller
  subsections and expanding brief paragraphs to improve flow and adherence to academic
  writing standards.

**Reviewer 2**

**General Comment**

- Reviewer Comment: "The manuscript reported mechanism of delayed surges in straits: seiche-induced oscillation triggered by typhoon passage. The issue is acceptable and method is along with scientific ways. But the manuscript does not reach the standard form to be published in the journal. Specially, proofreading is necessary by native English speakers. Details will be found below."
- Author's Response: We thank the reviewer for acknowledging the scientific merit of our study.
   We have undertaken a thorough revision of the manuscript to meet the journal's publication standards. As mentioned in our response to Reviewer 1, a native English speaker has proofread the entire manuscript to significantly improve its linguistic quality.

**Specific Comments**

**1. Abstract Clarity**

- Reviewer Comment: "The study's novelty is ambiguous. Also, what the authors did and what they found should be clarified."
- Author's Response: We have revised the Abstract to clearly describe the novelty, methodology, and key findings of the study. Specifically, we now highlight that this study is the first to identify the mechanism of delayed storm surges caused by westward-curving typhoons (WCTS-type) in strait regions, using wavelet analysis to extract 10- and 5-hour oscillations shown to be natural modes of the Tsushima Strait. Furthermore, we conducted numerical and idealized experiments to confirm that these oscillations amplify and delay storm surges. The revised Abstract now emphasizes the role of resonance effects and provides broader implications for other straits with similar geographic and meteorological conditions.
- Changes in Manuscript: The Abstract has been rewritten to explicitly state the study's novelty, methodology, key findings, and broader implications.

**2. Structural and Non-structural Measures**

- Reviewer Comment: "What are structural and non-structural measures for mitigating storm surge damage?"
- Author's Response: To address your point, we have revised the manuscript to clarify the distinction between structural and non-structural measures. Specifically, we now state that structural measures include storm surge barriers (e.g., Esteban et al., 2014), while non-

- structural measures include real-time forecasting systems (e.g., Igarashi et al., 2021). We have also emphasized that a comprehensive understanding of storm surge behavior is critical for designing and implementing effective mitigation strategies.
- Changes in Manuscript: Clarifications and examples of structural (e.g., storm surge barriers) and non-structural (e.g., real-time forecasting systems) storm surge mitigation measures have been added in lines 25-28. Moreover, because the surge can manifest with a significant delay, a premature sense of security following an initial decrease in water levels must be avoided. This necessitates the implementation of strategies that postpone the safe return time for evacuees in lines 377-382.

**3. Low Air Pressure Theory and Wind/Pressure Coincidence**

- Reviewer Comment: "What are the references for the theory of low air pressure? Why don't you think both wind and pressure coincidently work?"
- Author's Response: To clarify the theoretical basis for the effect of low air pressure on sea level, we have now cited Wunsch and Stammer (1997), who provide evidence for the inverse barometer effect in the context of large-scale sea level variations. Regarding the interaction of wind and pressure, we fully agree that both forces act simultaneously. However, our results indicate that their respective contributions to storm surge height are approximately linear and additive under the conditions examined. To explicitly demonstrate this, we added a new subsection entitled "Contributions of Pressure and Wind to Storm Surge Anomalies," in which we compare the storm surge responses from pressure-only, wind-only, and combined-forcing simulations. This comparison confirms that the combined effect closely matches the sum of the individual effects, supporting the assumption of linearity in this context.
- Changes in Manuscript: Wunsch and Stammer (1997) has been cited to support the inverse barometer effect. A new subsection, "Contributions of Pressure and Wind to Storm Surge Anomalies," has been added in section 5.3, presenting numerical experiments that demonstrate the linear and additive nature of wind and pressure contributions to storm surge.

**4. Definition of Wind Setup Height**

- Reviewer Comment: "In 40, what is wind setup height? On the head, the definition of wind setup should be given for better understanding."
- Author's Response: To improve clarity, we have added a definition of wind set-up in the revised manuscript. Specifically, we now state that: "wind setup refers to the rise in sea level due to strong winds forcing seawater shoreward (e.g., Walton and Dean (2009)). This occurs

- when coastal winds drive surface water toward the shore, causing accumulation in shallow regions and elevating the local sea level."
- Changes in Manuscript: A clear definition of "wind setup" has been added to the manuscript in lines 30-32.

**5. Clarification on Bays Facing the Ocean**

- Reviewer Comment: "In 40, 'Regions that have historically suffered significant storm surge damage are often bays facing the ocean, characterized by geographical conditions that amplify wind setup' -> What do you mean? All bays face the ocean, suffering from storm surges."
- Author's Response: We agree that the original sentence was ambiguous. To clarify our point, we have revised the text and added a new schematic figure titled "Schematic figure of typhoon positions and wind directions during its passage through the WCTS: (a) when the typhoon is located over the Tsushima Strait; (b) when the typhoon is located over the Sea of Japan." This figure illustrates how wind direction and coastline orientation interact during typhoon passage, showing that certain bays—such as those facing directly toward the prevailing wind direction—experience stronger wind set-up effects due to geographic alignment. We have also revised the explanation in the text to reflect this more clearly.
- Changes in Manuscript: The ambiguous sentence has been rephrased. A new schematic figure
  (Figure 2) has been added to visually explain how specific bay orientations amplify wind
  setup under certain typhoon conditions. The accompanying text has been revised for clarity
  in lines 53-64.

**6. Explicit Location of 100 cm Surges**

- Reviewer Comment: "In 45, the authors should explicitly point out where 100 cm surges occurred."
- Author's Response: To address this point, we have revised the manuscript to explicitly
  indicate the location where the 100 cm storm surge occurred. Specifically, we added the
  location of Gwangyang—the affected area—on the figure 2.
- Changes in Manuscript: The location of Gwangyang, where the 100 cm storm surge occurred, has been explicitly marked on the Figure 1b.

**7. Novelty and Originality in Introduction**

- Reviewer Comment: "In Introduction, what is the novelty? What is the originality?"
- Author's Response: To clarify the novelty and originality of our study, we have revised the Introduction to explicitly state our research objective and unique approach.

• Changes in Manuscript: The Introduction section has been revised to clearly articulate the novelty, originality, and specific research objectives of the study in lines 70-76.

**8. Clarification of Typhoon Tracks and Hakata Station**

- Reviewer Comment: "In 2.1, normally, we don't know what are 1,898 tracks and where is Hakata tidal station. So, I suggest adding 1,898 tracks, 599 tracks, and Hakata. You can remove one of the figures among Figs. 1, 2, and 3. Or you can make new figure put them on a figure."
- Author's Response: In response, we have added a new subsection entitled "Selected Typhoons" to clarify the process of typhoon selection. Additionally, we created a new figure titled "Tracks of selected typhoons in this study," which shows the full set of 1,898 typhoon tracks, the subset of 599 westward-curving tracks, and highlights the location of Hakata tidal station. This figure provides a comprehensive overview of the selection process and the geographical context, which we believe improves the reader's understanding of the study domain and methodology.
- Changes in Manuscript: A new section 3, "Selected Typhoons," has been added. A new figure
   3, "Tracks of selected typhoons in this study," now illustrates the 1,898 total tracks, the 599 westward-curving tracks, and the location of Hakata tidal station.

**9. Full Name of RDMDB**

- Reviewer Comment: "In 80, what is RDMDB? Full name is first given."
- Author's Response: To address this point, we have revised the manuscript to spell out the full
  name of the abbreviation "RDMDB" as Regional Delayed Mode Data Base upon its first
  appearance.
- Changes in Manuscript: The full name "Regional Delayed Mode Data Base" has been provided for "RDMDB" at its first mention in line 106.

**10. Wind Speed Limits for Drag Coefficient**

- Reviewer Comment: "In 115, did you consider wind speed limits for the wind drag coefficient? If you don't consider it, what is the reason? If you consider it, what happens in your results?"
- Author's Response: In this study, we adopted the wind drag coefficient formulation proposed by Large and Pond (1981), which is also used in Niimi et al. (2022), a validated reference for storm surge simulations in this region. Although some studies suggest that the drag coefficient saturates at high wind speeds (e.g., at a maximum value of approximately 0.0025), we did not

impose such a cap. The corresponding wind speed at which Cs reaches 0.0025 in our formulation is approximately 30.9 m/s. Since the maximum wind speeds in our target typhoons are around 35 m/s, we consider that applying a cap would have only a limited effect on the calculated storm surges. Therefore, the omission of a wind speed limit does not significantly affect our conclusions.

• Changes in Manuscript: Lines 185-186 now include the justification for consistency with Niimi et al. (2022).

**11. Organization of Section 2 Data and Methodology**

- Reviewer Comment: "I suggest editing Section 2 Data and Methodology. For instance, data, models, and simulation experiments. In particular, the description of simulation experiments is strongly suggested to look over the study."
- Author's Response: We have significantly revised the structure of the manuscript to clarify
  the methodology. Specifically, we reorganized the relevant content into separate sections for
  observational data, numerical model setup, and simulation experiments.
- Changes in Manuscript: The content of the former Section 2 has been reorganized into more
  distinct sections, including "Observed Storm Surge and weather data Characteristics,"
  "Numerical Simulation (with Model Setup and Validation)," and "Spectral Analysis using
  Wavelet Transform," to clearly delineate data, model, and experimental descriptions.

**12. Section 3 "Results and Discussion" Readability and Separation**

- Reviewer Comment: "Section 3 Results and discussion is unreadable because of too separated paragraphs in each section. The title of 3.3.1 might be better if it was like the effect of the superposition of oscillations on surges. My suggestion is that you can rename the titles of subsections with like impacts of something, not tools. It will be more readable. Why don't you absolutely separate the two sections of "Results" and "Discussion"? The reviewer cannot find discussions referring to other literature or comparing them to other works."
- Author's Response: We appreciate your insightful feedback regarding the organization of the
  results and discussion sections. Following your suggestion, we have completely restructured
  this part of the manuscript. The results are now divided into two main sections:
  - o Numerical Simulation, with impact-oriented subsections such as:
    - Model Setup
    - Validation
    - Contributions of Pressure and Wind to Storm Surge Anomalies (This specifically addresses your suggestion for impact-oriented titles.)

- o Spectral Analysis using Wavelet Transform, with revised titles like:
  - Continuous Wavelet Transform
  - Dominant Oscillation Modes Identified by CWT (These new titles are also crafted to be more descriptive of the scientific findings rather than simply the tools used.)

In addition, we created a dedicated Discussion section to interpret the physical mechanisms behind the delayed storm surges. This section also includes comparisons with previous studies in the strait regions. We believe this separation significantly improves the clarity and logical flow of the manuscript, and more effectively communicates the study's scientific contributions.

• Changes in Manuscript: The former "Results and Discussion" section has been entirely restructured. Results are now presented under two distinct main sections ("Numerical Simulation" and "Spectral Analysis using Wavelet Transform") with revised, impact-oriented subsection titles. A completely new, separate "Discussion" section has been created to interpret findings and compare them with existing literature, particularly for strait regions. This fundamental reorganization enhances clarity and logical flow.

**Editor**

**Detailed comments**

1.

- Comment: Lines 10-11. (i) I suggest ". . corresponding to natural modes . ." (omit "the" which tends to imply "all" natural modes). (ii) "(1, 1)" and "(1, 0)" are not explained here. It might be better to write ". . interpreted as low-mode seiches . ." here; the detail is in section 7.
- Author's Response: (i) As suggested, we have removed "the" and now describe the phenomenon as "corresponding to natural modes of the Tsushima Strait" to avoid implying that all natural modes are included. (ii) We agree that "(1, 1)" and "(1, 0)" are not clearly explained at this point in the manuscript. Therefore, we have revised the sentence to read "... interpreted as low-mode seiches ...," as suggested. The detailed discussion remains in Section 7.
- Changes in Manuscript: In line 10, we changed the text to "corresponding to natural modes of the Tsushima Strait", and in line 11, "interpreted as low-mode seiches"

2.

- Comment: Lines 28 and 441. ". . (Igarashi and Tajima (2021)). . ." Lines 67-69. Please put parentheses around these citations, e.g. ". . Strait (Zhang et al. (2010)), the . ."
- Author's Response: As suggested, we have removed the double parentheses. In addition, we have carefully revised other related parts in the manuscript.
- Changes in Manuscript:
  - In line 26: (Balaguru et al., 2016; Yang et al., 2020; Mori et al., 2022)
  - In line 27: (Miguel Esteban, 2014)
  - ➤ In line 28: (Igarashi and Tajima, 2021)
  - ➤ In line 30: (Wunsch and Stammer, 1997
  - In line 31: (Walton and Dean, 2009)
  - ➤ In line 34: (Wu et al., 2018).
  - In line 37: (Kim et al., 2010; Shen and Gong, 2009),
  - ➤ In line 38: (Kennedy et al., 2011).
  - ➤ In lines 41-42: (Bilskie et al., 2016; Bhaskaran et al., 2020; Nakajo et al., 2015; Ide et al., 2020).
  - In lines 47-48: (Yoon et al., 2014)
  - ➤ In line 2022).
  - ➤ In line 65: (Hong and Yoon, 1992; Niimi et al., 2022).

- In lines 66-67: (Zhang et al., 2010), the Strait of Georgia (Soontiens et al., 2016), and the Singapore Strait (Tkalich et al., 2013),
- In line 134: (Hong and Yoon, 1992).
- ➤ In line 162: (Yoon and Shim, 2013).
- In line 163: (Mellor and Yamada, 1982; Smagorinsky, 1963)
- In line 182: (Large and Pond, 1981)
- > 274: (Lee et al., 2019)

3.

- Comment: Line 176. ".. in Chen et al. (2003)." Is sufficient.
- Author's Response: As suggested, we changed the text.
- Changes in Manuscript: In line 174: "... procedures is available in Chen et al. (2003)."

4.

- Comment: Line 325. Better to begin "In (18) (20) we let the coastline . ."
- Author's Response: As suggested, we changed the text.
- Changes in Manuscript: In line 323: "In Equations (18) (20), we let the coastline"

5.

- Comment: Lines 411-412. Data availability. This statement may be OK regarding model code but please refer to the "Statement on the availability of underlying data" at https://www.ocean-science.net/policies/data\_policy.html . "The best way to provide access to data is by depositing them . . in . . reliable public data repositories." "The data needed to replicate figures in a paper should in any case be publicly available . ."
- Author's Response: The datasets and files analyzed during the current study are available from
  the corresponding author upon reasonable request. However, some input data were obtained from
  third-party commercial sources and are not publicly accessible due to licensing restrictions. These
  proprietary datasets are described in the manuscript, and information on how to access them is
  available from the authors.
- Changes in Manuscript: In lines 410-412, we changed the statement to "The datasets and files analyzed during the current study are available from the corresponding author upon reasonable request. However, some input data were obtained from third-party commercial sources and are not publicly accessible due to licensing restrictions. These proprietary datasets are described in the manuscript, and information on how to access them is available from the authors."

We hope that the revisions and clarifications outlined above adequately address all reviewer and editor concerns. Please do not hesitate to contact us should any additional information be required.